# Signaling Peptide SpoV Is Essential for *Streptococcus pyogenes* Virulence, and Prophylaxis with Anti-SpoV Decreases Disease Severity

**DOI:** 10.3390/microorganisms9112321

**Published:** 2021-11-10

**Authors:** Andrea L. Herrera, Michael S. Chaussee

**Affiliations:** Division of Basic Biomedical Sciences, The Sanford School of Medicine, The University of South Dakota, Vermillion, SD 57069, USA; Andrea.Herrera@usd.edu

**Keywords:** *Streptococcus pyogenes*, SpoV, virulence, iGAS, antibody therapy

## Abstract

Streptococcal peptide of virulence (SpoV) is a *Streptococcus pyogenes* (group A streptococcus (GAS))-specific peptide that is important for GAS survival in murine blood, and the expression of the virulence factors streptolysin O (*slo*) and streptolysin S (*sagA*). We used a *spoV* mutant in isolate MGAS315 to assess the contribution of the SpoV peptide to virulence by using a murine model of invasive disease and an ex vivo human model (Lancefield assay). We then used antibodies to SpoV in both models to evaluate their ability to decrease morbidity and mortality. Results showed that SpoV is essential for GAS virulence, and targeting the peptide has therapeutic potential.

## 1. Introduction

*Streptococcus pyogenes* (group A streptococcus (GAS)) is an exclusively human pathogen that causes both mild and invasive multiple diseases. Mild infections of the throat and skin include pharyngitis and impetigo. Life-threatening invasive GAS (iGAS) diseases include bacteremia, streptococcal toxic shock syndrome, pneumonia, and necrotizing fasciitis (“flesh-eating disease”). iGAS diseases are significantly concerning because they have high mortality rates despite the availability of antibiotics that are effective ex vivo [1,2]. The diversity and severity of GAS diseases is partly attributed to the pathogens’ ability to regulate the expression of a variety of virulence factors, including adherence and invasion proteins, toxins, superantigens, proteases, and immune-modulating proteins [3]. Consequently, to cause disease, GAS must be able to adapt to and grow in many different environments within the human host.

GAS uses extracellular peptides as signaling molecules to regulate the expression of virulence genes [4,5]. Propeptides are synthesized and then post-translationally processed during secretion to biologically active extracellular signaling peptides. Extracellular peptides can be detected either at the cell surface or intracellularly [5]. Peptides are typically detected at the cell surface by a membrane-bound sensor kinase. The sensor kinase responds by transferring a phosphoryl group to a response regulator protein to change its DNA-binding specificity, which results in the activation or repression of target genes. Alternatively, peptides can be actively transported into the cell, where the peptide can directly interact with a transcriptional regulator to alter target gene expression [6,7,8]. Several characterized GAS signaling peptides influence pathogenesis by utilizing both mechanisms [9,10,11,12,13,14,15].

We previously identified the streptococcal peptide of virulence (SpoV) in culture supernatants of MGAS315 when screening for GAS signaling peptides [16]. A BLASTP search of the National Center for Biotechnology Information (NCBI) database using SpyM3_0132 as a query identified 1982 similar sequences among GAS isolates. We performed signal peptide cleavage site predictions for SpoV using SignalP 5.0 [16]. The software predicted that, in isolate MGAS315, SpoV contains a typical bacterial signal peptide of 31 amino acids followed by a secreted 20 amino acid extracellular peptide [16]. The extracellular 20 amino acid SpoV peptide (NDASFYGHTGPDSWLLYTVW) is found among 7% of sequenced GAS isolates, and there is no amino acid sequence variation among GAS isolates that encode the 20 amino acid extracellular SpoV [16]. The majority (93%) of GAS isolates encode a 55 amino acid peptide, which is processed to an extracellular 24 amino acid SpoV peptide [16]. Thirteen different amino acid sequence variations of the 24 amino acid SpoV peptide occur among the 1982 GAS isolates identified in our BLASTP search [16]. The main difference between the 20 and 24 amino acid extracellular SpoV peptides is the presence or absence of amino acids tyrosine, serine, asparagine, and glycine (YSNG) near the N terminus. While our analysis was limited, gene expression was equally affected following the addition of either the 20 or 24 amino acid peptides, indicating that both peptide variants have the same effect on GAS gene expression [16].

The expression of *spoV* varies among GAS isolates due to allelic variation in *rocA* (regulator of CovS), which is a component of the control of virulence (CovRS) regulatory system [16]. Mutations to *covS* can naturally occur during infection, which alters the transcription of CovR regulated genes such as *slo* and results in more invasive GAS diseases [17,18]. SpoV is also important for the expression of several CovRS regulated genes, including *slo*, *sagA* (streptolysin S; SLS), and *speB* (streptococcal exotoxin B); however, the direct mechanisms involved in the SpoV-mediated gene regulation of CovRS-regulated genes are unknown [16]. One way in which pore-forming toxins SLO and SLS are associated with iGAS disease is by forming large pores in host cell membranes, which disrupts their integrity [19,20]. The virulence of SpeB throughout infection is complex. SpeB cleaves multiple host proteins, including extracellular matrix proteins, immunoglobulins, and antimicrobial peptides [21,22], which interferes with host immune functions. Additionally, SpeB cleaves several GAS proteins, including the M protein [23], superantigens [24,25], and streptokinase [26], which interferes their functions. Changes in virulence gene expression suggest that SpoV is likely to be important for GAS virulence.

SpoV is not encoded in the genomes of any other bacterial species, but orthologs are present in the genomes of all GAS isolates. In all GAS isolates, SpoV is encoded proximal to the *slo* gene, which encodes the SLO cytolysin. The deletion of *spoV* decreased SLO-specific hemolytic activity and resistance to murine immune effector cells [16]. Further, the deletion of *spoV* and subsequent addition of synthesized SpoV peptides increased *slo* expression [16]. Because peptide signaling plays an important regulatory role during disease progression, and SpoV affects virulence gene expression, we hypothesized that SpoV may contribute to GAS virulence. In this study, the contribution of SpoV to GAS virulence, and the efficacy of anti-SpoV immunotherapy are evaluated.

## 2. Materials and Methods

### 2.1. Strain and Culture Conditions

Frozen stocks of GAS isolate MGAS315 (serotype M3) (American Type Culture Collection; ATCC; Manassas, VA, USA) were stored in 50% glycerol at −80 °C until use. Frozen stocks were streaked onto Todd-Hewitt (THY) (Becton Dickinson; Sparks, MD, USA) agar and incubated overnight at 37 °C in a 5% CO_2_ atmosphere. Then, 10 mL of THY broth containing 0.2% (*w*/*v*) yeast extract (Becton Dickinson) was used to suspend GAS in liquid media without shaking. The culture was then grown at 37 °C in a 5% CO_2_ atmosphere. *Escherichia coli* was grown with a Luria–Bertani (LB) (Fisher Scientific; Waltham, MA, USA) medium at 37 °C with agitation or on LB agar plates. When appropriate, spectinomycin (Spec; 100 µg/mL for both *S. pyogenes* and *E. coli*), or chloramphenicol (Cm; 10 µg/mL for *S. pyogenes* and 50 µg/mL for *E. coli*) was added to the media. Antibiotics were purchased from Fisher Scientific; Waltham, MA, USA.

### 2.2. DNA Manipulation

Standard protocols or manufacturer’s instructions were used to isolate plasmid DNA, and conduct restriction endonuclease, DNA ligase, PCR, and other enzymatic treatments of plasmids and DNA fragments. Enzymes were purchased from New England Biolabs, Inc. (NEB; Ipswich, MA, USA). Phusion DNA polymerase (NEB; Ipswich, MA, USA) was used to amplify specific regions of the MGAS315 chromosome. Oligonucleotides were purchased from Eurofins Genomics (Louisville, KY, USA).

### 2.3. Construction of spoV Mutant

The MGAS315 *spoV* deletion mutant was constructed by double-crossover recombination using a plasmid (pAH3) that contained a spectinomycin-resistance gene (SpecR; *aad9*) cloned between DNA identical to DNA sequences located upstream and downstream of *spoV* [16].

### 2.4. Construction of spoV Complemented Mutant

We complemented the MGAS315 *spoV* mutant by using a derivative of the pAM401 shuttle vector with *spoV* expression controlled by the native *spoV* promoter. The *spoV* ORF and the nucleotide sequence beginning 183 bases upstream of the *spoV* start codon were amplified from the MGAS315 genome with PCR, and cloned into the pAM401 vector using restriction sites SalI and BamHI, which were present within the primers. Recombinant plasmid pAH5 was then used to transform the MGAS315 *spoV* mutant by electroporation, and transformants were selected on agar plates containing chloramphenicol.

### 2.5. Generation of Anti-SpoV Antibodies

Four rabbits were each immunized with a total of 4 mg of a synthesized SpoV (ANDASFYGQNAPDSWLLYTV) peptide (Biomatik; Cambridge, Ontario, Canada). The 1st and 2nd immunizations (350 µg antigen/rabbit) used complete Freund’s adjuvant and were administered on days 0 and 14. The 3rd–8th immunizations (150 µg antigen/rabbit) used incomplete Freund’s adjuvant and were administered on days 28, 35, 42, 49, 56, and 64. Blood was harvested on day 72, and anti-SpoV antibodies were antigen-specific affinity-purified.

### 2.6. Ex Vivo Human Model of Virulence

MGAS315 wt, the *spoV* mutant, or the *spoV* complemented strain were grown with THY broth at 37 °C to an *A*_600_ of 0.6–0.8, centrifuged, suspended in PBS, aliquoted, and stored at −80 °C as frozen stocks. The number of viable GAS cells in the stock preparation was enumerated by dilution plating with THY agar plates. A human blood sample (BioIVT; Baltimore, MD, USA) from pooled donors (male and female) was pretreated with IdeS (100 U/mL) (Sigma-Aldrich; St. Louis, MO, USA) for 1 h at 37 °C [27]. Then, GAS stocks were thawed on ice, and approximately 1000 CFUs (diluted in 15 µL PBS) were added to 270 µL of whole human blood and incubated at 37 °C with rotation. Diluted blood samples were plated onto agar plates after 3 h to enumerate the surviving bacteria.

In some experiments, MGAS315 wt was incubated with 6 µg of anti-SpoV, normal rabbit IgG (rb-IgG) (Sigma-Aldrich; St. Louis, MO, USA), or PBS for 15 min at room temperature. After exposure to anti-SpoV or controls, bacteria were mixed with 270 µL human blood (not pretreated with IdeS) and incubated at 37 °C for 3 h. The survival rate was calculated as (CFU (at a given time point)/CFU (at the start)) × 100.

### 2.7. Mice

Female Balb/c mice were purchased from Jackson Laboratories (Indianapolis, IN, USA) and used in experiments when they were between 6 and 8 weeks of age. They were housed in cages and given 24 h access to food and water. All animal experiments were conducted in accordance with the recommendations in the Guide for the Care and Use of Laboratory Animals of the National Institutes of Health and according to the guidelines of the local Institutional Animal Care and Use Committee of the University of South Dakota. Protocol number 10-06-18-21E was approved on 25 July 2018. Victor Huber and Ruth Bakker provided all animal care and handling training. All efforts were made to minimize suffering, ensure the highest ethical standard, and adhere to the 3R principle (reduction, refinement, and replacement).

### 2.8. Murine Model of iGAS Disease

GAS was harvested during the mid-exponential phase of growth and diluted 1 × 10^5^ CFUs/500 μL. Mice were lightly anesthetized with 2.5% isoflurane, and GAS was given intraperitoneally (i.p.) using a 25-gauge needle, which was inserted at approximately 45° into the side of the abdominal wall. LD_50_ was determined by the method of Reed and Muench [28]. Animal health and behavior was monitored at least three times a day over the 14 day course of the experiment. Body weight was recorded daily. Endpoint criteria included extreme clinical signs of infection (huddling, hunched posture, ruffled fur, tachypnea), severe hypothermia as indicated by a temperature of 34 °C (~4.5 °C below normal), and weight loss equal to or greater than 20% of starting weight. Mice with one or more of these symptoms were immediately euthanized, and the infection was considered to be lethal. 

### 2.9. Passive Immunotherapy with Anti-SpoV Antibodies

In some experiments, 0.5 mg of anti-SpoV, rb-IgG, or PBS (500 μL) was administered two hours prior to GAS infection. Antibodies were administered i.p. 

### 2.10. GAS Quantification in Mouse Tissues

Mouse tissue samples were collected 24 h after GAS inoculation. Blood was collected from the submandibular vein of each animal, and immediately after they had been euthanized by CO_2_ inhalation, the lungs and spleens were removed and homogenized in sterile PBS. GAS were enumerated by dilution plating with THY or blood agar plates as previously described [29].

### 2.11. Statistics

All quantification and statistical data analyses were conducted with GraphPad Prism 8 software (San Diego, CA, USA). Unless otherwise stated, error bars represent standard error of the mean (SEM), and *p* values were calculated using either one-way analysis of variance (ANOVA) with a Tukey multiple-comparison post hoc test or a Student’s *t*-test. For mouse survival studies, results were graphed as Kaplan–Meier curves, and data were analyzed using the log-rank test. Values were accepted as significant if the *p* value was less than 0.05.

## 3. Results

### 3.1. Assessment of Virulence in a Murine Model of Invasive GAS Disease

To determine if SpoV is important for GAS virulence, we compared morbidity and mortality in a murine model of iGAS disease. Groups of mice were inoculated intraperitoneally (i.p) with the MGAS315 wt strain, the isogenic *spoV* mutant, or the *spoV* complemented mutant, and mortality was assessed for 14 days after infection (Figure 1). Of mice infected with the *spoV* mutant, 100% (12/12) survived, whereas only 17% (2/12) mice infected with MGAS315 wt survived the infection. Overall, mortality was significantly greater in mice inoculated with the parental MGAS315 strain compared to mice inoculated with the *spoV* mutant (*p <* 0.0001). There was also significantly greater morbidity (weight loss) with MGAS315 wt compared to the *spoV* mutant (Appendix A). We complemented the *spoV* mutant by transforming it with a shuttle plasmid that expressed the *spoV* open reading frame adjacent to its promotor. Complementation decreased survival compared to the *spoV* mutant, and survival was like that observed with wild-type infection (*p* < 0.0001). Overall, results showed that SpoV is essential for virulence in a mouse model of invasive infection.

### 3.2. SpoV Enhanced GAS Survival in an Ex Vivo Human Model of Virulence

Invasive GAS diseases are often associated with bacteremia. Therefore, the ability of GAS to survive and grow in whole human blood was tested using an ex vivo human model of virulence (Lancefield assay). Blood was pretreated with IdeS to degrade IgG that might be specific to MGAS315 [30]. Then, MGAS315 wt, the isogenic *spoV* mutant, or the *spoV* complemented mutant was added and incubated at 37 °C for 3 h prior to determining the number of viable bacteria by dilution plating. Fewer CFUs were recovered from blood containing the MGAS315 *spoV* mutant compared to blood containing either the MGAS315 wt strain or the complemented mutant strain (Figure 2; *p* < 0.05). Overall, results showed that SpoV enhances GAS virulence in an ex vivo human model of virulence.

### 3.3. SpoV Decreased GAS Dissemination in a Mouse Model of Systemic Infection

Growth in blood mimics a key feature of invasive GAS diseases, which is the ability to disseminate and potentially colonize multiple organs. To investigate whether SpoV could affect GAS dissemination, and hence contribute to virulence, we compared the number of viable GAS recovered from the blood, spleen, and lungs of mice 24 h after i.p. inoculation with MGAS315 wt, the *spoV* mutant, or the *spoV* complemented strain (Figure 3). There were fewer GAS recovered from the blood of mice infected with the *spoV* mutant compared to mice infected with the wt strain (*p* < 0.05). Specifically, we did not culture any GAS from the blood of mice inoculated with the *spoV* mutant. Compared to the *spoV* mutant, a greater number (10^3^ CFU/mL) of CFUs were recovered in the blood following infection with the *spoV* complemented mutant; however, the difference was not statistically significant.

GAS enumeration in the spleen showed that there was a 1.4 log decrease in the number of CFUs among mice inoculated with the *spoV* mutant compared to mice infected with the MGAS315 wt strain, and complementation of the mutant partially increased the number of GAS recovered in the spleen, but differences were not statistically significant. Enumeration of GAS in the lungs showed that 50% (2/4) of the mice injected with the *spoV* mutant did not have recoverable GAS in their lungs, whereas 100% (4/4) of mice infected with MGAS315 wt or the complemented *spoV* mutant did, although results were again not statistically significant. Overall, deletion of *spoV* decreased the number of bacteria in the blood, and while the results were not statistically significant, this also resulted in fewer CFUs recovered from the spleen and lungs following i.p infection.

### 3.4. Administration of Anti-SpoV Decreased GAS Survival in an Ex Vivo Human Model of Virulence

To determine if antigen-purified antibodies specific to SpoV (anti-SpoV) could decrease GAS survival in human blood, approximately 1000 CFUs of MGAS315 wt, the *spoV* mutant, or the *spoV* complemented strains were preincubated with anti-SpoV, a rabbit control IgG (rb-IgG), or PBS and mixed with human blood at 37 °C. After 3 h, diluted samples were plated onto THY agar plates for enumeration (Figure 4). In this experiment, the human blood was not pretreated with IdeS to ensure that neither anti-SpoV nor rb-IgG were degraded. Treatment with anti-SpoV significantly enhanced the killing of MGAS315 wt (red circles) in human blood compared to the addition of rb-IgG (gray circles) or PBS (black circles). Treatment with anti-SpoV had no effect on the killing of *spoV* mutant (red squares) compared to the addition of rb-IgG (gray squares; *p* < 0.05) or PBS (black squares; *p* < 0.05). Treatment with anti-SpoV had no effect on the killing of *spoV* mutant (red squares) compared to the addition of rb-IgG (gray squares) but enhanced GAS killing when samples were treated with PBS (black squares; *p* < 0.05). The *spoV* complemented strain (red triangles) had significantly increased resistance to killing in whole blood compared to the *spoV* mutant (red squares; *p* < 0.05), indicating a specificity of anti-SpoV to SpoV.

### 3.5. Prophylactic Protection following Addition of Anti-spoV in a Murine Model of iGAS Disease

Next, we tested the protective efficacy of anti-SpoV in a mouse model of iGAS disease. Mice were inoculated i.p. with 0.5 mg anti-SpoV, rb-IgG, or PBS. Two hours later, mice were inoculated with MGAS315 wt. Two independent experiments were completed. After 14 days, 28% of the animals that had received PBS and 44% of mice that had received rb-IgG survived the GAS infection (Figure 5a). In contrast, 60% of the mice that had received anti-SpoV survived iGAS infection. Differences in survival among the groups were not statistically different.

Overall, mice showed the greatest survival against iGAS when they were prophylactically treated with anti-SpoV; however, nonimmune rabbit IgG (rb-IgG) also increased survival compared to mice that received no treatment (PBS). Anti-SpoV treatment also prolonged death from iGAS disease compared to both control-treated groups of mice. Specifically, of the mice that succumbed to infection, the last death was recorded 9 days after treatment with anti-SpoV, compared to 6 or 2 days after mice had been treated with rb-IgG or PBS, respectively. Additionally, morbidity was significantly decreased in mice that had received anti-SpoV treatment compared to mice that had received PBS (*p* < 0.0001) or rb-IgG (*p* < 0.001) (Figure 5b). Mice that had received PBS never regained their initial starting weight, whereas mice that had received anti-SpoV recovered to 100% of their initial starting weight approximately 5 days after infection, which is a striking effect on morbidity.

## 4. Discussion

GAS secretes peptides that can function as signaling molecules to influence the production of extracellular virulence factors [9,10,13,14] and biofilm production [11,12]. Additionally, gene expression within the competence regulon is activated by a peptide [15]. GAS peptide SpoV is important for the expression of several genes associated with virulence, including *slo*, *sagA* (streptolysin S), and *speB* [16]. In this study, we used a *spoV* mutant to assess the contribution of SpoV to GAS virulence by using a murine model of invasive disease and an ex vivo human model (Lancefield assay). We then used anti-SpoV antibodies in both models to evaluate their therapeutic potential. Results showed that SpoV is essential for GAS virulence, and targeting the peptide could mitigate severe GAS disease.

Prophylactic treatment of mice with anti-SpoV resulted in increased survival compared to treatment with either rb-IgG or PBS. However, treatment with rb-IgG also increased survival compared to in untreated mice (PBS) (Figure 5a). GAS expresses several surface-bound proteins that bind to nonimmune immunoglobulins (Igs) via the Fc region, which interferes with Fc-mediated phagocytosis by immune cells. In addition to human IgG, GAS binds nonspecifically bind IgG from several animal species, including rabbits [31]. The M protein and M-like proteins bind to both IgG and IgA via the Fc region, which inhibits phagocytosis and increases GAS survival in blood [31,32,33]. SfbI binds to the Fc region of IgG, which prevents antibody-dependent cell cytotoxicity (ADCC) by macrophages [34]. Protein H interacts with Fc region IgG and inhibits IgG-dependent complement activation [35]. Overall, nonimmune IgG binding may contribute to some increased protection compared to mice treated with PBS (Figure 5).

Small peptide antagonists were examined in GAS as a way of disrupting peptide-signaling systems [36,37]. In GAS, short hydrophobic peptides 2/3 (Shp2 and Shp3) induce biofilm production by interacting directly with the transcriptional regulators Rgg2 and Rgg3, respectively [12]. Aggarwal et al. used a high-throughput screen to identify compounds that specifically disrupted Shp–Rgg3 complexes and thereby interfered with Rgg3-regulated pathways, including biofilm development [36]. The addition of peptide antagonists blocked Shp-dependent biofilm formation, indicating that they can disrupt GAS peptide signaling [36]. Thus, peptide antagonists have shown promise at disrupting peptide signaling in GAS; however, the small compounds must be imported into the cytoplasm and compete for binding to the transcriptional regulator in order to exert their function.

To date, there are no reports of antibodies targeting GAS signaling peptides to disrupt their function. In *Staphylococcus aureus*, the production of several virulence factors, including pore-forming toxin α-hemolysin, are controlled by the accessory gene regulator (*agr*) quorum-sensing (QS) system [38]. Similar to GAS peptide signaling systems that use linear peptides, the *agr* system utilizes small cyclic peptides, termed autoinducing peptides (AIPs), as its signaling molecules [39]. Park et al. generated a monoclonal antibody (AP4-24H11) against AIP-4 and showed that AP4-24H11 reduced α-hemolysin expression and activity when supernatants were treated with AP4-24H11 [40]. Additionally, a simultaneous injection of 10^8^ *S. aureus* with 0.6 mg AP4-24H11 prevented abscess formation in a subcutaneous infection model, compared to controls [40]. In our study, we tested a single 0.5 mg i.p. injection of polyclonal antibodies to SpoV two hours prior to a lethal GAS infection, which significantly decreased morbidity and moderately decreased mortality compared to the controls (Figure 5). Park et al. also used an infection model similar to our study: a single 1 mg i.p. injection of AP4-24H1 given two hours prior to a lethal *S. aureus* infection. Prophylactic treatment with AP4-24H11 resulted in 100% protection compared to control-treated mice [40]. Thus, targeting an AIP with an antibody was sufficient to suppress peptide signaling and virulence in *S. aureus.*

SpoV signaling directly contributes to GAS pathogenesis by controlling the expression of several important virulence determinants, such as toxins, proteases, and other immune-evading factors [16]. Peptide signaling relies on coordinating virulence gene expression among a bacterial population. Thus, blocking SpoV-mediated signaling with anti-SpoV presumably causes GAS to lose the ability express critical virulence factors necessary to mount an organized defense against the host immune system. Because SpoV is found in all sequenced GAS isolates, and no orthologs have been detected among other species, including other streptococcal species, SpoV appears to be unique to GAS [16]. Therefore, anti-SpoV is unlikely to disrupt the normal flora. Overall, the development of GAS-specific antibodies, such as those targeting SpoV, may be useful as an adjunct treatment to decrease the morbidity and mortality of iGAS diseases.

## Figures and Tables

**Figure 1 microorganisms-09-02321-f001:**
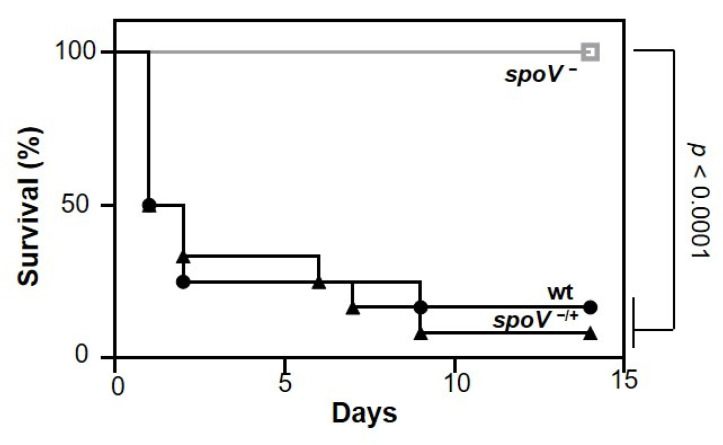
SpoV is essential for virulence in a murine model of iGAS disease. Kaplan–Meier survival analysis shown for mice following intraperitoneal infection (1 × 10^5^ CFU) with MGAS315 wt, *spoV* mutant (*spoV^−^*), or *spoV* complemented mutant (*spoV^−^*^/+^). Mortality was assessed for 14 days after infection (n = 12) (*p*  <  0.0001 by log-rank test).

**Figure 2 microorganisms-09-02321-f002:**
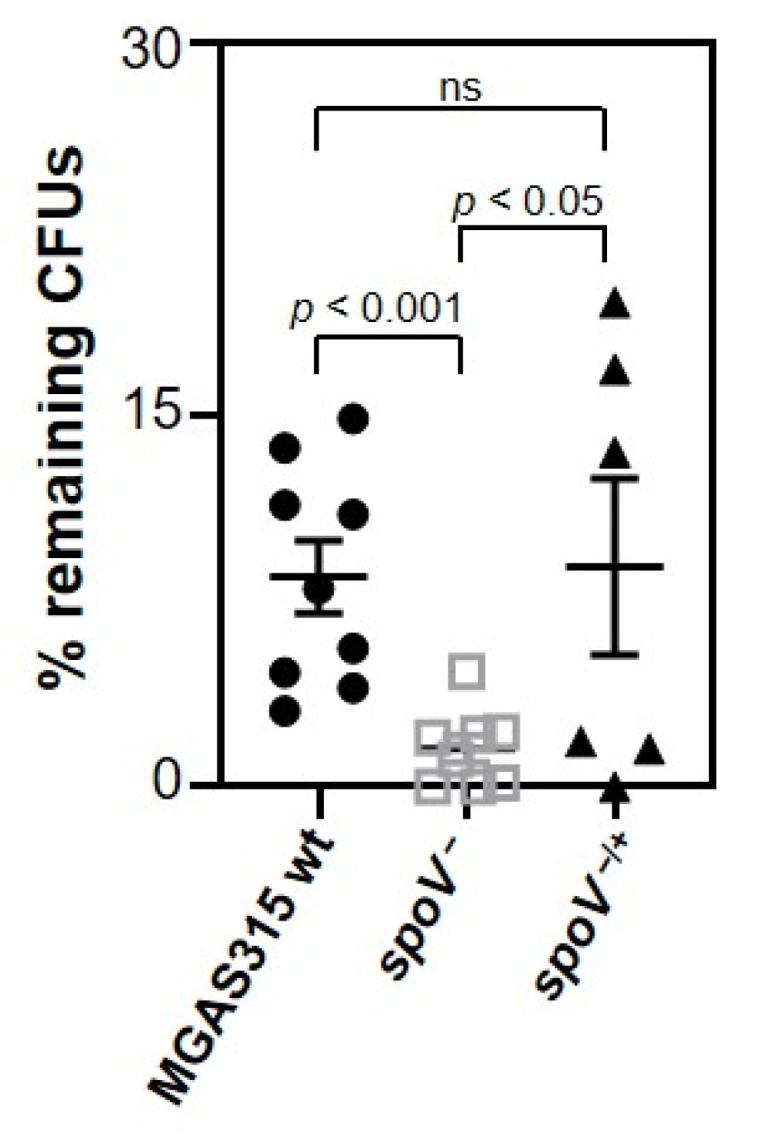
SpoV is essential for virulence in an ex vivo human model of virulence. Human blood from pooled donors (270 µL) was treated with 100 U/mL IdeS. Approximately 1000 CFUs of each bacterial strain were added and incubated at 37 °C. After 3 h, diluted samples were plated onto THY agar plates for enumeration. Three different human blood collections, each consisting of one female and one male donor, were used for independent experiments that were completed in triplicate or duplicate. Lines represent the mean and SEM value in each group. Statistical significance was determined by using one-way ANOVA with Tukey’s multiple-comparison test. *p* values determined by comparison among the indicated strains.

**Figure 3 microorganisms-09-02321-f003:**
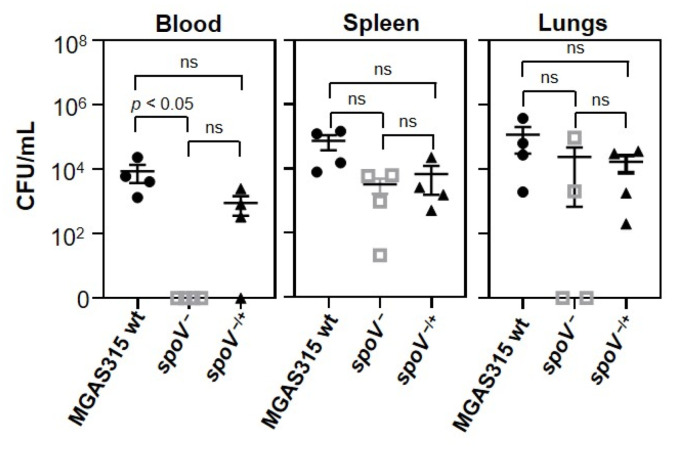
Deletion of *spoV* decreased GAS dissemination in a murine model of iGAS disease. Dissemination of GAS determined 24 h following intraperitoneal infection (1 × 10^5^ CFU) with MGAS315 wt, the *spoV* mutant, or the *spoV* complemented mutant (n = 4). Mice were euthanized 24 h after GAS infection, and total number of CFUs in blood, spleen, and lungs was determined for each mouse. Lines represent the mean and SEM value in each group. Significance determined by using one-way ANOVA with Tukey’s multiple-comparison test (ns, not significant).

**Figure 4 microorganisms-09-02321-f004:**
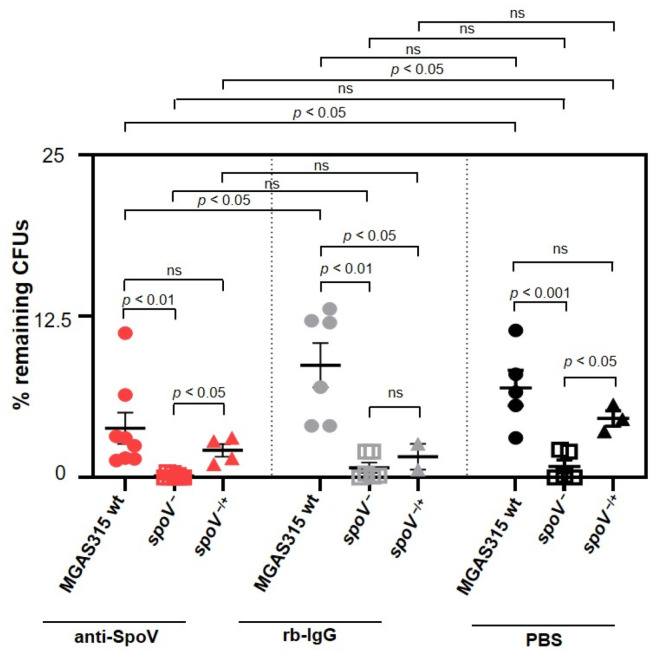
Anti-SpoV decreased MGAS315 wt survival in an ex vivo model of virulence. MGAS315 wt (circles), the *spoV* mutant (squares), or *spoV* complemented (triangle) strains incubated with anti-SpoV (red), control rabbit IgG (rb-IgG) (gray), or PBS (black) for 15 min at room temperature; then, 270 µL of pooled whole human blood added and incubated at 37 °C. After 3 h, viable CFUs were enumerated by dilution plating. Three different human blood collections, each consisting of one female and one male donor, were used for independent experiments that were completed in triplicate or duplicate. Lines represent the mean and SEM value in each group. Significance assessed by using one-way ANOVA with Tukey’s multiple-comparison test (ns, not significant).

**Figure 5 microorganisms-09-02321-f005:**
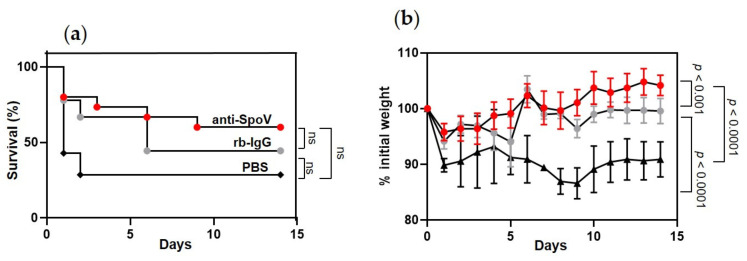
Prophylactic administration of anti-SpoV increased GAS survival against infection with MGAS315 wt. (**a**) Kaplan–Meier survival analysis for groups of mice injected i.p. with 0.5 mg of anti-SpoV, control rabbit IgG (rb-IgG), or PBS 2 h prior to i.p. infection of MGAS315 wt (1 × 10^5^ CFUs). Mortality assessed for 14 days after infection from two independent experiments (n = 15). (**b**) Morbidity (weight loss) assessed for 14 days after infection from two independent experiments (n = 15; ns, not significant).

## Data Availability

Not applicable.

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
