# Peer review of "Signaling Peptide SpoV Is Essential for Streptococcus pyogenes Virulence, and Prophylaxis with Anti-SpoV Decreases Disease Severity"

_microorganisms, 2021, doi:10.3390/microorganisms9112321_

Round 1

Reviewer 1 Report

Herrera et al present a paper assessing the contribution of a small signaling peptide (SpoV) on group A streptococcal (GAS) virulence.  The authors build on their earlier study showing SpoV’s importance in regulating the virulence genes in GAS, to demonstrate its importance in infection models. They first show a significant decrease in mortality when infecting mice with GAS mutants that lack SpoV. Mice infected with these mutants also display significantly reduced levels of GAS in their blood. Furthermore, GAS strains lacking SpoV show a decreased ability to survive in human blood. As the authors have thoroughly demonstrated SpoV’s contributions to virulence, the authors create antibodies against SpoV to treat GAS infections. These antibodies are used to treat mice prophylactically, and are able to provide a degree of protection against infection. Although this work lacks direct mechanistic insight, combined with previous work, this work clearly establishes the importance of SpoV in modulating virulence in GAS in multiple infection models.

Major Points:

Figure 4 and 5 demonstrate the ability of anti-SpoV antibodies to decrease GAS replication/virulence in human and mouse models. In figure 5, it seems that adding the control rabbit (rb-IgG) also has quite significant protective effects. This should not be ignored and should be mentioned in the text along with a possible explanation. In addition, to strengthen the claim that anti-SpoV antibodies are providing specific protection, it would be useful to repeat these experiments with the SpoV knockout strain. Though repeating the mouse experiments may be needlessly laborious, repeating the experiments with the mutant in Figure 4 should be doable.  

Minor Points:

The significance between Wt and MGAS315 and the complemented mutant should be shown

Line 213, typo in statistically

The discussion and/or introduction would benefit from an expansion on the mechanism of SpoV virulence. Adding how the mentioned factors (slo, sagA, speB) effect GAS virulence would give better background for the in vivo work carried out in this study.

Author Response

Reviewer 1

Major Points:

Figure 4 and 5 demonstrate the ability of anti-SpoV antibodies to decrease GAS replication/virulence in human and mouse models. In figure 5, it seems that adding the control rabbit (rb-IgG) also has quite significant protective effects. This should not be ignored and should be mentioned in the text along with a possible explanation. In addition, to strengthen the claim that anti-SpoV antibodies are providing specific protection, it would be useful to repeat these experiments with the SpoV knockout strain. Though repeating the mouse experiments may be needlessly laborious, repeating the experiments with the mutant in Figure 4 should be doable.  

GAS can bind to immunoglobulin non-specifically via the Fc region, which may increase resistance to phagocytosis. We included this explanation in the text.  (Line 354-365)

As requested, we included the spoV mutant and complemented strains as well as untreated (PBS) controls to Figure 4. The spoV mutant strain had increased killing in whole blood compared to the spoV complemented strain indicating a specificity of the antibody to SpoV. The results were discussed in the text.  (Line 286-287, 292, 293-299, 304, 305; updated Figure 4)

We appreciate the suggestion of adding additional studies using the spoV mutant as a control to test the effect with anti-SpoV in our animal model (Figure 5). We agree that including these experiments would be too laborsome for this manuscript. This is in part because determining an inoculum size for the spoV knockout that causes death (such that we could detect a protective effect of anti-SpoV) would likely require many additional mice. Also, using large inoculums presents interpretation problems in these models.

Minor Points:

The significance between Wt and MGAS315 and the complemented mutant should be shown

Statistical significance is now shown on all figures between all groups. (Updated Figure 4 and 5)

Line 213, typo in statistically

We corrected the grammatical error. (Line 261)

The discussion and/or introduction would benefit from an expansion on the mechanism of SpoV virulence. Adding how the mentioned factors (slo, sagA, speB) effect GAS virulence would give better background for the in vivo work carried out in this study.

We added more information about SpoV virulence, including that SpoV is part of the major regulatory network of CovRS which is involved in controlling several important virulence genes. Additionally, we added details about how slo, sagA, and speB affect GAS virulence. (Line 68-70, 72-81)

Reviewer 2 Report

Minor points:

Line 120: "[...] onto agar plates at after 3 hours [...]" --> remove "at"

Fig. 2: IdeS treatment of the blood samples should be mentioned in the figure caption, as it is not self-explanatory

lines 206: Remove period before (Figure 3). 

207-209: Somewhat awkwardly worded.

Line 224: "[...] of CFUs among of mice [...]" --> remove "of"

Line 229-231: If the difference is not statistically significant, it should not be claimed that there was a decreased dissemination to spleen and lungs. This is a tendency at max.

Fig. 5: Signifcance levels should be indicated in the Kaplan-Meyer plot. How can the effect of the rb-IgG explained?

Major Point:

Figure 4: Control of untreated GAS should be included.

Author Response

Reviewer 2

Minor points:

Line 120: "[...] onto agar plates at after 3 hours [...]" --> remove "at"

We corrected the grammatical error. (Line 147)

Fig. 2: IdeS treatment of the blood samples should be mentioned in the figure caption, as it is not self-explanatory

We included a statement that blood samples were treated with 100 U/ml of IdeS to the figure ledged. (Line 242)

lines 206: Remove period before (Figure 3). 

          The period was removed. (Line 255)

207-209: Somewhat awkwardly worded.

          We re-worded this statement. (Line 240-241)

Line 224: "[...] of CFUs among of mice [...]" --> remove "of"

          We removed “of” (Line 273)

Line 229-231: If the difference is not statistically significant, it should not be claimed that there was a decreased dissemination to spleen and lungs. This is a tendency at max.

We removed statements about GAS dissemination to the spleen and lungs. There were lower number of CFUs in the spoV mutant compared to the wt GAS, however the difference was not statistically significant, and is stated in the text. (Line 281)

Fig. 5: Significance levels should be indicated in the Kaplan-Meyer plot. How can the effect of the rb-IgG explained?

As requested, statistical significance is now shown on all figures between all groups. GAS can bind to rb-IgG non-specifically which may cause some increased resistance to phagocytosis. We included this information in the text.  (Line 339, 352-363; updated Figure 5)

Major Point:

Figure 4: Control of untreated GAS should be included.

Growth in human blood with untreated (PBS) GAS was added to Figure 4 and results were discussed in the text. (Line 293-297; updated Figure 4)

Reviewer 3 Report

The manuscript by Herrera and Chaussee demonstrates the role of the SpoV signaling peptide from Streptococcus pyogenes in virulence and dissemination. In addition, use of anti-SpoV resulted in attenuated infection in the WT GAS strain. Overall, the manuscript is well written and presents the data in a logical format. There are just a few comments below that would improve the presentation.

  1. The figure legend of figure 1 states the inoculum for the mouse infection, but this needs to be stated in the materials and methods too.
  2. Line 58 -59 states that there was no difference in “regulation of gene expression” upon the addition of SpoV peptide. But, lines 68-69 state that when SpoV peptide was added there was increased expression of slo. Are you saying in the first sentences that the addition of the peptide did not impact the gene expression of just spoV? This needs to be clarified.
  3. What was the p-value for the mortality in Figure 5a? P-values or “ns” are shown for every other piece of data.
  4. The discussion section could be shortened – the detailing of experiments from other studies used in support of their findings could be written much more concisely.

Author Response

Reviewer 3

  1. The figure legend of figure 1 states the inoculum for the mouse infection, but this needs to be stated in the materials and methods too.

Mice were infected with 105 CFUS and this was added to the materials and methods section. (Line 170)

  1. Line 58 -59 states that there was no difference in “regulation of gene expression” upon the addition of SpoV peptide. But, lines 68-69 state that when SpoV peptide was added there was increased expression of slo. Are you saying in the first sentences that the addition of the peptide did not impact the gene expression of just spoV? This needs to be clarified.

We clarified the statements about prior experiments with addition of synthetic peptides. There was no difference to GAS gene expression when either C21 or C25 were added to cultures. Each peptide variant was equally able to increase, decrease, or cause no change to GAS gene expression. (Line 61-62, 63-64)

  1. What was the p-value for the mortality in Figure 5a? P-values or “ns” are shown for every other piece of data.

Statistical significance is now shown on all figures between all groups and stated in the text. (Line 319-320, 339; updated Figure 5)

  1. The discussion section could be shortened – the detailing of experiments from other studies used in support of their findings could be written much more concisely.

In order to shorten the discussion section, we deleted some details of other studies which we used to support our findings. (Line 370-373, 375-377)

Additional changes:

We re-worded our Material and methods section to decrease similarity to our previously published manuscripts (Line 94-98, 99-104, 155-165, 168-171, 172-173, 184-186). We also added our animal protocol number (project identification code), date of approval and name of the ethics committee and institutional review board to the Material and Methods section as requested (Line 156-165, 424-430).

Round 2

Reviewer 2 Report

The authors have addressed all concerns.